# Evaluation of Quality of Life and Emotional Disturbances in Patients with Diabetic Retinopathy

**George Saitakis** [1,*]**, Dimitrios Roukas** [2] **, Erifili Hatziagelaki** [3]**, Vasiliki Efstathiou** [4]**, Panagiotis Theodossiadis** [1] **and Emmanouil Rizos** [4]

[1] Second Department of Ophthalmology, National and Kapodistrian University of Athens, 'Attikon' University General Hospital, 12462 Athens, Greece
[2] Department of Psychiatry, 417 VA (NIMITS) Hospital, 11521 Athens, Greece; droukas@hotmail.gr
[3] Research Institute and Diabetes Center, Second Department of Internal Medicine-Propaedeutic, National and Kapodistrian University of Athens, 'Attikon' University General Hospital, 12462 Athens, Greece
[4] Second Department of Psychiatry, National and Kapodistrian University of Athens, Medical School of Athens, 'Attikon' University General Hospital, 12462 Athens, Greece; vefstathiou@psych.uoa.gr (V.E.); erizos@med.uoa.gr (E.R.)
[*] Correspondence: giorgos.saitakis@gmail.com; Tel.: +30-6978487844

**Abstract:** Diabetes has detrimental effects on many organs, including the kidneys, heart, and the central nervous system, with ophthalmic involvement and Diabetic Retinopathy (DR), specifically, being among the most severe and prominent consequences. Diabetic Retinopathy and especially advanced stages of the disease, have a crucial impact on patients' quality of life and emotional status. In this context, emotional imbalance, psychological side effects and comorbidities, like anxiety disorders, could emerge, deteriorating the patients' condition further. A number of questionnaires can be employed in the evaluation of the potential impact of Diabetic Retinopathy on patients' quality of life, including the Beck Anxiety Inventory (BAI) and The National Eye Institute Visual Function Questionnaire-25 (NEI VFQ-25). Purpose: The purpose of this study was to evaluate the association of Diabetic Retinopathy (DR) and diabetic macular edema with vision-related quality of life, as well as the potential association between the disease's severity, emotional status of patients and the manifestation of anxiety and psychological features. Results: Patients with fundoscopic findings had significantly lower scores in all VFQ-25 subscales, indicating worse quality of life in comparison to patients without DR. Severity of DR, greater levels of anxiety, daily sitting time, unemployment and lower education level, were all found to be significantly, negatively associated with a worse quality of life. Regarding emotional status, more years of suffering from diabetes, treatment with insulin and the hours being idle per day were associated with an increased burden of anxiety. In addition, the presence of a concomitant disease, findings in fundoscopy, diabetic macular edema and treatment with anti-VEFG injections, as well as the number of doses, were significantly associated with greater anxiety. Multivariate analysis showed that having Severe Non-Proliferative Diabetic Retinopathy or having Proliferative Diabetic Retinopathy and receiving insulin therapy (alone or in combination with another treatment), were significantly associated with higher levels of anxiety. Conclusion: The well-established impact of DR on the patients' well-being, quality of life and emotional status render DR and CME prevention, stabilization or delaying progression as a necessity in order to protect patients from developing psychiatric symptoms. On the other hand, the speculated bi-directional association between emotional problems and DR progression highlights the importance of acknowledging and dealing with psychological issues with the aim of delaying DR progression.

**Keywords:** evaluation of life in diabetes; emotional disturbances in Diabetic Retinopathy; emotional status in Diabetic Retinopathy; anxiety in Diabetic Retinopathy

## 1. Introduction

Diabetes mellitus (DM) represents one of the major morbidities worldwide. It constitutes a chronic, multisystemic, metabolic disorder, characterized by increased glucose serum levels [1–3]. DM has detrimental effects on many organs, including the kidneys, heart, and central nervous system, with ophthalmic involvement and Diabetic Retinopathy (DR), specifically, being among the most severe and prominent consequences [4–6]. A number of inhibitory and stimulating angiogenetic factors are involved in the development of DR, with Vascular Endothelial Growth Factor (VEGF) playing a crucial role [7–10]. DR is clinically separated into two main categories based on the disease's severity and the presence or absence of abnormal retinal neovascularization, i.e., Non-Proliferative DR and Proliferative DR [3,7]. Major causes of severe visual impairment in DR patients include: (a) diabetic macular edema (DME) secondary to capillary leakage, (b) macular ischemia and optic nerve involvement (diabetic papillopathy), caused by capillary occlusion, and (c) hemorrhages, tractional retinal detachment, and neovascular glaucoma, in the context of retinal neovascularization [4,11–15].

The incidence of DR associated with chronicity of the disease and aging, is increased irrespective of DR type. The hallmark study in this field was the Wisconsin Epidemiologic Study of Diabetic Retinopathy (WESDR), which demonstrated that after 20 years of DR, almost 99% of Type 1 and 60% of Type 2 patients would have developed some degree of Diabetic Retinopathy. In general, the vast majority of patients suffering from DR-Type 1, and nearly 2 out of 3 patients with DR-Type 2, are expected to have developed retinal disease of varying severity a decade after the initial diagnosis [16,17].

Diabetic Retinopathy and especially the advanced stages of the disease, have a crucial impact on patients' quality of life and emotional status. Visually impaired patients, secondary to DR, face difficulties in many fundamental aspects of their daily life and have to deal with significant burdens. In this context, psychological distress as well as psychological disorders, such as anxiety disorders and even depression, could emerge, further deteriorating patients' condition [18–20]. Anxiety represents the most frequently encountered mental disorder in Europe, characterized clinically by alertness and feelings of fear, tension and anxiety in dealing with daily life events [21–24]. A number of questionnaires can be employed in the evaluation of the potential impact of Diabetic Retinopathy on patients' quality of life. Patient reported outcomes are crucial in order for the burden of the disease to be estimated.

To the best of our knowledge, little information is provided in the literature regarding evidence-based reasoning about the degree of social and emotional impact of DR and the subsequent necessity of psychological support to those diagnosed with this condition.

The aim of this study is to assess the quality of life in patients with DR or DR including diabetic macular edema, providing a better understanding of the psychological impact of DR. It also aims to investigate the potential association between the disease's severity, emotional status of patients, and the manifestation of anxiety and psychological disorders, utilizing the BAI and VFQ scales.

## 2. Materials and Methods

This study included 142 patients suffering from diabetes mellitus, who were examined at the retina service of the 2nd Department of Ophthalmology, "Attikon" University Hospital of Athens, Greece. This study was in adherence with the Tenets of the Declaration of Helsinki and was approved by the institutional review board. Informed consent was obtained from all patients before they were included in this study.

The sample included both DM-Type 1 and DM-Type 2 diabetics who were recruited either at their initial presentation at the diabetic retinopathy service or had been examined previously at our department and agreed to participate in this study at one of their regular follow-ups. All participants had their Best Corrected Visual Acuity measured by the Snellen chart [25]. In addition, they underwent slit-lamp biomicroscopy and dilated fundoscopy to be evaluated for the presence or absence of Diabetic Retinopathy. Diabetics without DR

findings constituted the control group (Group I). All participants who presented with DR were staged by an ophthalmologist, according to the Early Treatment Diabetic Retinopathy Study (ETDRS) criteria, as non-proliferative mild DR (Group II), moderate DR (Group III), severe DR (Group IV) and proliferative DR (Group V) [26]. For each patient the stage was determined based on the eye depicting the most severe retinal lesions. Furthermore, presence of diabetic macular edema was evaluated utilizing optical coherence tomography (Stratus OCT3, Carl Zeiss, Germany). Additionally, previous treatment with anti-VEGF agents was recorded, as well as the number of injections given in each eye, whenever this data was available.

Social and demographic data were recorded for all participants, including age, gender, family and employment status, educational level (primary school at most, middle/high school, college/university/MSc), type and duration of diabetes mellitus, current treatment (pills or/and insulin), as well as the presence of other comorbidities, such as hypertension, high cholesterol, heart/thyroid/kidneys disease, and previous events of cerebral or myocardial infarctions. In addition, participants responded to questions regarding smoking habits, alcohol consumption, engagement in physical activities and time spent idle per day.

All participants were requested to complete two self-report questionnaires, the first one reflecting their anxiety status, the Beck Anxiety Inventory, and the second one, which is the National Eye Institute Visual Function Questionnaire-25, evaluates the vision-related quality of life [27–29]. The BAI is a psychometric rating scale used to evaluate the severity of anxiety symptoms, developed by Aaron T Beck, MD. It contains 21 self-report items reflecting symptoms of anxiety, including numbness or tingling, feeling hot, wobbliness in legs, ability to relax, fear of the worst happening, dizziness or lightheadedness, pounding or racing heart, unsteadiness, feeling terrified, feeling nervous, feeling of choking, hand tremors, feeling shaky, fear of losing control, difficulty breathing, fear of dying, feeling scared, indigestion or abdominal discomfort, faintness, face flushing and sweating. Each item allows the patient four choices ranging from no symptoms to severe symptoms. For each item, the patient is asked to report how he or she has felt during the past week. The scores are classified as minimal anxiety (0 to 7), mild anxiety (8 to 15), moderate anxiety (16 to 25) and severe anxiety (30 to 63). The questionnaire can be given to the same patient in subsequent sessions to track the progression or improvement of the anxiety. The BAI discriminates effectively between anxious and non-anxious diagnostic groups and, as a result, it is useful as a screening measure for anxiety. The reliability coefficient is 0.92. The test–retest reliability is 0.75 [30]. Correlations of the BAI with a set of self-report and clinician-rated scales were all significant (e.g., Spearman rank correlation coefficient ($r_s$) > 0.50 [27–29,31,32].

The NEI VFQ-25 is one of the most commonly used instruments in the evaluation of vision-related QOL as it meets the required criteria of measuring a number of crucial qualitative features, including the impact of vision on everyday activities, emotional well-being, social relationships, and independence. In addition, it addresses the three components recommended by the World Health Organization's International Classification of Functioning Disability and Health (WHO-ICF) for measuring health-related consequences of a disease, that is, impairment, activity limitations, and participation restriction. Globally, the NEI-VFQ-25 has been translated into several languages, including Greek [33], with slightly varying degrees of reliability and validity and has been utilized in several population-based studies. The NEI-VFQ was originally developed with 51 items (13 subscales) to measure patients' self-reported vision-dependent function and the impact of vision problems on health-related quality of life (QOL) across several common eye conditions. NEI-VFQ-25 represents a shortened version, consisting of 25 questionnaire items, which has been demonstrated to be internally consistent, reproducible, responsive and more appropriate for use in daily clinical practice. Compared to its predecessor, it includes all the subscales except for the "expectation" one. The 12 subscales (25 items) that were included, evaluate the following dimensions of vision-targeted, health-related quality of life: overall general health (1); overall general vision (1); difficulties with near (3), distance (3), peripheral (1), and color vision (1); vision-specific role difficulties (2), dependency on others

due to diminished vision (3), mental health problems (4), and limitations to social function due to visual impairment (2); ocular pain (2); and driving difficulties (2). Each of the subscales has a score from 0 to 100, in which 100 indicates the best possible and 0 the worst possible function.

Initially, we evaluated the impact of DR on the BAI and VF-25 questionnaires based on the existence (first arm) or the absence (second arm) of funduscopic findings. Consequentially, subgroup analysis was performed, comparing the findings in the anxiety and quality of life scales among the 4 groups of different DR severity stages [29,34,35].

*Statistical Analysis*

Quantitative variables were expressed as the mean (standard deviation) or as the median (interquartile range). Qualitative variables were expressed as absolute and relative frequencies. Mann–Whitney tests were used for the comparison of continuous variables between the two groups. Multiple linear regression analyses were used with dependent variables, all VFQ-25 scales, and BAI scales in a stepwise method ($p$ for entry 0.05, $p$ for removal 0.10). Adjusted regression coefficients ($\beta$) with standard errors (SE) were computed from the results of the linear regression analyses. Logarithmic transformations of the dependent variables were used in the linear regression analyses. All reported $p$ values are two-tailed. Statistical significance was set at $p < 0.05$ and analyses were conducted using SPSS statistical software (version 22.0).

## 3. Results

The sample consisted of 142 patients (53.5% women) with a mean age of 59.4 years (SD = 17.4 years). Patients' characteristics are presented in Table 1. The majority of the participants (63.4%) were married or living with their partner. A total of 38.7% of the sample was employed while 35.9% were primary school graduates at most. Current smokers constituted 20.4% of the participants. One out of four (25.4%) was physically active and 21.3% were consuming alcohol. Most of the participants had Type 2 diabetes (62.7%), suffered from a concomitant disease (84.5%) and had fundoscopy findings (69.0%). More specifically, 27.5% of the patients had Proliferative Diabetic Retinopathy and 16.2% had mild Non-Proliferative Diabetic Retinopathy.

**Table 1.** Sample characteristics.

|  | N (%) |
|---|---|
| Gender |  |
| Men | 66 (46.5) |
| Women | 76 (53.5) |
| Age, mean (SD) | 59.4 (17.4) |
| Married/Living with partner | 90 (63.4) |
| Employed | 55 (38.7) |
| Educational level |  |
| Primary school at most | 51 (35.9) |
| Middle/High school | 48 (33.8) |
| College/University/MSc | 43 (30.3) |
| Current smoker | 29 (20.4) |
| Ex-smoker | 29 (20.4) |
| Alcohol consumption | 30 (21.3) |
| Physically active | 36 (25.4) |
| Time spending sitting per day (hours), median (IQR) | 6 (4–8) |
| Years with diabetes, mean (SD) | 18.8 (9.4) |
| Type of diabetes |  |
| Type 1 | 53 (37.3) |
| Type 2 | 89 (62.7) |
| Treatment |  |
| Pills only or combined with something else | 78 (54.9) |
| Insulin only or combined with something else | 84 (59.2) |
| Concomitant disease | 120 (84.5) |

**Table 1.** *Cont.*

|  | N (%) |
|---|---|
| **Fundoscopy** |  |
| Without findings | 44 (31.0) |
| Mild Non-Proliferative Diabetic Retinopathy | 23 (16.2) |
| Moderate Non-Proliferative Diabetic Retinopathy | 18 (12.7) |
| Severe Non-Proliferative Diabetic Retinopathy | 18 (12.7) |
| Productive Diabetic Retinopathy | 39 (27.5) |
| Diabetic Macular Edema | 46 (32.4) |
| Treatment with anti-VEFG injections | 34 (24.1) |
| Number of doses, median (IQR) | 9 (5–12) |
| Beck Anxiety Index, median (IQR) | 22.5 (9–34) |

Participants' scores in all VFQ-25 subscales are presented in Table 2. Patients with fundoscopy findings had significantly lower scores in all subscales, as well as in the composite score, indicating worse quality of life than patients without findings in fundoscopy.

**Table 2.** Participants' scores in all VFQ-25 subscales.

|  | **Total Sample** | | **Fundoscopy** | | | | *p* Mann–Whitney Test |
|---|---|---|---|---|---|---|---|
|  |  |  | **Without Findings** | | **With Findings** | |  |
|  | **Mean (SD)** | **Median (IQR)** | **Mean (SD)** | **Median (IQR)** | **Mean (SD)** | **Median (IQR)** |  |
| General Health | 40.7 (30.3) | 25 (25–50) | 63.1 (29.3) | 62.5 (50–100) | 30.6 (25) | 25 (25–50) | <0.001 |
| General Vision | 64.1 (24.5) | 60 (40–80) | 81.4 (17.5) | 80 (60–100) | 56.3 (23.2) | 60 (40–80) | <0.001 |
| Ocular Pain | 68.6 (27.3) | 75 (50–100) | 84.4 (19.1) | 93.8 (75–100) | 61.5 (27.5) | 62.5 (37.5–75) | <0.001 |
| Near Activities | 67.6 (29.2) | 66.7 (41.7–100) | 87.1 (21.6) | 100 (83.3–100) | 58.8 (28) | 58.3 (33.3–83.3) | <0.001 |
| Distance Activities | 67.8 (32.9) | 83.3 (33.3–100) | 89.4 (19.7) | 100 (83.3–100) | 58.2 (33.1) | 54.2 (25–91.7) | <0.001 |
| Social Functioning | 77.8 (29.8) | 100 (62.5–100) | 95.7 (11.7) | 100 (100–100) | 69.7 (31.9) | 75 (37.5–100) | <0.001 |
| Mental Health | 65.1 (32.1) | 75 (37.5–93.8) | 87.6 (17.3) | 93.8 (81.3–100) | 54.9 (32) | 62.5 (25–81.3) | <0.001 |
| Role Difficulties | 66.8 (33.7) | 75 (50–100) | 86.9 (24.7) | 100 (87.5–100) | 57.8 (33.4) | 50 (25–87.5) | <0.001 |
| Dependency | 73.1 (33.3) | 87.5 (50–100) | 93.9 (17) | 100 (100–100) | 63.7 (34.6) | 75 (33.3–100) | <0.001 |
| Driving | 75.6 (28.5) | 87.5 (58.3–100) | 86.2 (25.5) | 100 (83.3–100) | 67.7 (28.4) | 58.3 (50–100) | 0.010 |
| Color Vision | 79.9 (27.3) | 100 (50–100) | 93.8 (16.3) | 100 (100–100) | 73.7 (29) | 75 (50–100) | <0.001 |
| Peripheral Vision | 76.6 (28.7) | 100 (50–100) | 93.8 (14.4) | 100 (100–100) | 68.8 (30.2) | 75 (50–100) | <0.001 |
| Composite VFQ-25 score | 67.8 (26.6) | 73.8 (49.6–91.6) | 86.7 (15.4) | 92.8 (82.3–97.9) | 59.3 (26.2) | 61.6 (36.2–81.2) | <0.001 |

Notes: Participants' scores in all VFQ-25 subscales and their association with having findings in fundoscopy; VFQ: Visual Function Questionary.

When multiple linear regression was conducted it was found that the results of the fundoscopy were significantly and independently associated with all VFQ-25 subscales (except for driving) as well as the composite score (Table 3). More specifically, patients with Moderate Non-Proliferative Diabetic Retinopathy, Severe Non-Proliferative Diabetic Retinopathy or Proliferative Diabetic Retinopathy had significantly worse general health compared to patients without findings in their fundoscopy. In the rest of the subscales and in the composite score, the significant difference was found between patients with Proliferative Diabetic Retinopathy and those without findings in their fundoscopy, with the patients with Proliferative Diabetic Retinopathy having worse quality of life. Greater anxiety symptoms were significantly associated with worse quality of life, in all sectors except for general health, driving and color vision. Daily idle time was significantly associated in a negative way with general health, general vision, distance activities, social functioning, mental health and composite VFQ-25 score. Employed patients had significantly higher scores in near activities, distance activities, social functioning, mental health, dependency and peripheral vision scales, indicating better quality of life. Finally, patients who were primary school graduates at most had significantly worse general vision and significantly lower composite VFQ-25 score compared to patients who were college/university or MSc graduates.

**Table 3.** Multivariate regression analyses results with scores in VFQ-25 subscales as dependent variables.

| | | B + | SE ++ | p |
|---|---|---|---|---|
| **General Health** | | | | |
| Fundoscopy | Without findings (reference) | | | |
| | Mild Non-Proliferative Diabetic Retinopathy | −0.201 | 0.144 | 0.166 |
| | Moderate Non-Proliferative Diabetic Retinopathy | −0.315 | 0.157 | **0.047** |
| | Severe Non-Proliferative Diabetic Retinopathy | −0.376 | 0.159 | **0.019** |
| | Proliferative Diabetic Retinopathy | −0.712 | 0.127 | **<0.001** |
| | Time spending sitting per day (hours) | −0.051 | 0.014 | **<0.001** |
| **General Vision** | | | | |
| Educational level | College/University/MSc (reference) | | | |
| | Primary school at most | −0.120 | 0.048 | **0.014** |
| | Middle/High school | −0.057 | 0.047 | 0.228 |
| Fundoscopy | Without findings (reference) | | | |
| | Mild Non-Proliferative Diabetic Retinopathy | −0.014 | 0.057 | 0.801 |
| | Moderate Non-Proliferative Diabetic Retinopathy | −0.029 | 0.063 | 0.648 |
| | Severe Non-Proliferative Diabetic Retinopathy | −0.043 | 0.065 | 0.510 |
| | Proliferative Diabetic Retinopathy | −0.255 | 0.055 | **<0.001** |
| | Time spending sitting per day (hours) | −0.020 | 0.006 | **0.001** |
| | Beck Anxiety Score | −0.004 | 0.001 | **0.002** |
| **Ocular Pain** | | | | |
| Fundoscopy | Without findings (reference) | | | |
| | Mild Non-Proliferative Diabetic Retinopathy | −0.084 | 0.073 | 0.253 |
| | Moderate Non-Proliferative Diabetic Retinopathy | −0.050 | 0.080 | 0.530 |
| | Severe Non-Proliferative Diabetic Retinopathy | −0.018 | 0.082 | 0.823 |
| | Proliferative Diabetic Retinopathy | −0.270 | 0.066 | **<0.001** |
| | Beck Anxiety Score | −0.007 | 0.001 | **<0.001** |
| **Near Activities** | | | | |
| Employed | No (reference) | | | |
| | Yes | 0.147 | 0.053 | **0.007** |
| Fundoscopy | Without findings (reference) | | | |
| | Mild Non-Proliferative Diabetic Retinopathy | −0.086 | 0.078 | 0.270 |
| | Moderate Non-Proliferative Diabetic Retinopathy | −0.039 | 0.086 | 0.650 |
| | Severe Non-Proliferative Diabetic Retinopathy | −0.026 | 0.088 | 0.769 |
| | Proliferative Diabetic Retinopathy | −0.318 | 0.071 | **<0.001** |
| | Beck Anxiety Score | −0.003 | 0.002 | **0.050** |
| **Distance Activities** | | | | |
| Employed | No (reference) | | | |
| | Yes | 0.156 | 0.055 | **0.006** |
| Fundoscopy | Without findings (reference) | | | |
| | Mild Non-Proliferative Diabetic Retinopathy | −0.080 | 0.081 | 0.323 |
| | Moderate Non-Proliferative Diabetic Retinopathy | −0.155 | 0.088 | 0.083 |
| | Severe Non-Proliferative Diabetic Retinopathy | −0.032 | 0.091 | 0.729 |
| | Proliferative Diabetic Retinopathy | −0.361 | 0.075 | **<0.001** |
| | Time spending sitting per day (hours) | −0.016 | 0.008 | **0.047** |
| | Beck Anxiety Score | −0.005 | 0.002 | **0.002** |
| **Social Functioning** | | | | |
| Employed | No (reference) | | | |
| | Yes | 0.093 | 0.043 | **0.032** |
| Fundoscopy | Without findings (reference) | | | |
| | Mild Non-Proliferative Diabetic Retinopathy | −0.045 | 0.062 | 0.472 |
| | Moderate Non-Proliferative Diabetic Retinopathy | −0.023 | 0.069 | 0.741 |
| | Severe Non-Proliferative Diabetic Retinopathy | 0.020 | 0.070 | 0.779 |
| | Proliferative Diabetic Retinopathy | −0.290 | 0.057 | **<0.001** |
| | Time spending sitting per day (hours) | −0.027 | 0.006 | **<0.001** |
| | Beck Anxiety Score | −0.003 | 0.001 | **0.007** |
| | | β+ | SE++ | p |
| **Mental Health** | | | | |
| Employed | No (reference) | | | |
| | Yes | 0.144 | 0.055 | **0.009** |
| Fundoscopy | Without findings (reference) | | | |
| | Mild Non-Proliferative Diabetic Retinopathy | −0.124 | 0.079 | 0.121 |
| | Moderate Non-Proliferative Diabetic Retinopathy | −0.028 | 0.087 | 0.750 |
| | Severe Non-Proliferative Diabetic Retinopathy | 0.017 | 0.090 | 0.852 |
| | Proliferative Diabetic Retinopathy | −0.357 | 0.074 | **<0.001** |
| | Time spending sitting per day (hours) | −0.029 | 0.008 | **0.001** |
| | Beck Anxiety Score | −0.010 | 0.002 | **<0.001** |
| **Role Difficulties** | | | | |
| Fundoscopy | Without findings (reference) | | | |
| | Mild Non-Proliferative Diabetic Retinopathy | −0.072 | 0.119 | 0.548 |
| | Moderate Non-Proliferative Diabetic Retinopathy | −0.007 | 0.131 | 0.958 |
| | Severe Non-Proliferative Diabetic Retinopathy | −0.060 | 0.133 | 0.655 |
| | Proliferative Diabetic Retinopathy | −0.463 | 0.108 | **<0.001** |
| | Beck Anxiety Score | −0.010 | 0.002 | **<0.001** |

**Table 3.** *Cont.*

|  |  | B + | SE ++ | p |
|---|---|---|---|---|
| Employed | No (reference) |  |  |  |
|  | Yes | 0.165 | 0.076 | **0.031** |
| Fundoscopy | Without findings (reference) |  |  |  |
|  | Mild Non-Proliferative Diabetic Retinopathy | −0.136 | 0.111 | 0.222 |
|  | Moderate Non-Proliferative Diabetic Retinopathy | 0.018 | 0.121 | 0.879 |
|  | Severe Non-Proliferative Diabetic Retinopathy | −0.012 | 0.124 | 0.923 |
|  | Proliferative Diabetic Retinopathy | −0.454 | 0.101 | **<0.001** |
|  | Beck Anxiety Score | −0.012 | 0.002 | **<0.001** |
| **Driving** |  |  |  |  |
| Age |  | −0.005 | 0.002 | **0.030** |
| **Color Vision** |  |  |  |  |
| Fundoscopy | Without findings (reference) |  |  |  |
|  | Mild Non-Proliferative Diabetic Retinopathy | −0.024 | 0.074 | 0.744 |
|  | Moderate Non-Proliferative Diabetic Retinopathy | −0.022 | 0.081 | 0.782 |
|  | Severe Non-Proliferative Diabetic Retinopathy | −0.082 | 0.081 | 0.308 |
|  | Proliferative Diabetic Retinopathy | −0.366 | 0.063 | **<0.001** |
| **Peripheral Vision** |  |  |  |  |
| Employed | No (reference) |  |  |  |
|  | Yes | 0.111 | 0.049 | **0.024** |
| Fundoscopy | Without findings (reference) |  |  |  |
|  | Mild Non-Proliferative Diabetic Retinopathy | −0.046 | 0.071 | 0.516 |
|  | Moderate Non-Proliferative Diabetic Retinopathy | −0.017 | 0.080 | 0.834 |
|  | Severe Non-Proliferative Diabetic Retinopathy | −0.053 | 0.080 | 0.505 |
|  | Proliferative Diabetic Retinopathy | −0.306 | 0.065 | **<0.001** |
|  | Beck Anxiety Score | −0.005 | 0.001 | **<0.001** |
| **Composite VFQ-25 score** |  |  |  |  |
| Educational level | College/University/MSc (reference) |  |  |  |
|  | Primary school at most | −0.119 | 0.042 | **0.005** |
|  | Middle/High school | −0.060 | 0.041 | 0.151 |
| Fundoscopy | Without findings (reference) |  |  |  |
|  | Mild Non-Proliferative Diabetic Retinopathy | −0.042 | 0.050 | 0.400 |
|  | Moderate Non-Proliferative Diabetic Retinopathy | −0.023 | 0.055 | 0.679 |
|  | Severe Non-Proliferative Diabetic Retinopathy | −0.006 | 0.057 | 0.920 |
|  | Proliferative Diabetic Retinopathy | −0.247 | 0.048 | **<0.001** |
|  | Time spending sitting per day (hours) | −0.022 | 0.005 | **<0.001** |
|  | Beck Anxiety Score | −0.005 | 0.001 | **<0.001** |

Note. Logarithmic transformations were used in these analyses. + regression coefficient ++ Standard Error.

The Beck Anxiety Index ranged from 0 to 63, with mean being 23.3 (SD = 17.7) and median being 22.5 (IQR: 9–34). Its association with participants' characteristics is presented in Table 4. A longer disease duration was significantly associated with higher levels of anxiety. Also, being treated with insulin (alone or in combined treatment) was significantly associated with greater anxiety. The longer the participants remained idle per day, the greater the anxiety they felt. Moreover, patients with concomitant disease, those with findings in fundoscopy, those who had diabetic macular edema and those who were under treatment with anti-VEFG injections, had significantly greater anxiety symptoms. In addition, more doses of anti-VEFG injections were significantly associated with greater anxiety.

**Table 4.** Association of BAI score with patients' characteristics.

|  | Beck Anxiety Score | | p |
|---|---|---|---|
|  | **Mean (SD)** | **Median (IQR)** | **Mann–Whitney Test** |
| Gender |  |  |  |
| Men | 24.5 (18.8) | 24.5 (10–36) | 0.611 |
| Women | 22.4 (16.7) | 21.5 (7.5–33.5) |  |
| Age, r ‡ | −0.09 |  | 0.297 |
| Married/Living with partner |  |  |  |
| No | 24.4 (18.3) | 22 (11–36.5) | 0.701 |
| Yes | 22.7 (17.4) | 22.5 (6–33) |  |
| Educational level |  |  |  |
| Primary school at most | 23.6 (19) | 21 (8–37) |  |
| Middle/High school | 22.8 (17.8) | 25 (4.5–35) | 0.982+ |
| College/University/MSc | 23.6 (16.3) | 24 (12–34) |  |
| Employed |  |  |  |
| No | 24.3 (19.1) | 22 (8–36) | 0.631 |
| Yes | 21.9 (15.3) | 23 (9–32) |  |

**Table 4.** *Cont.*

| | Beck Anxiety Score | | *p* |
|---|---|---|---|
| | **Mean (SD)** | **Median (IQR)** | **Mann–Whitney Test** |
| Smoking habits | | | |
| Smokers | 25.1 (20.2) | 26 (10–39) | |
| Nonsmokers | 23.2 (17.1) | 21 (9–34.5) | 0.827+ |
| Ex-smokers | 22 (17.4) | 23 (8–30) | |
| Alcohol consumption | | | |
| No | 23.7 (18) | 23 (9–36) | 0.789 |
| Yes | 22.6 (16.5) | 23.5 (10–30) | |
| Physically active | | | |
| No | 24.6 (18.4) | 25 (10–36) | 0.210 |
| Yes | 19.6 (15.1) | 20.5 (6–31) | |
| Time spending sitting per day (hours), r ‡ | 0.23 | | 0.007 |
| Years with diabetes, mean (SD) | 0.29 | | 0.001 |
| Type of diabetes | | | |
| Type 1 | 25.9 (17.1) | 25 (15–34) | 0.199 |
| Type 2 | 21.8 (17.9) | 21 (5–34) | |
| Treatment with pills only or combined with something else | | | |
| Pills only | 25.6 (16.4) | 25 (14.5–35.5) | 0.132 |
| Combined | 21.5 (18.5) | 19.5 (2–34) | |
| Treatment with insulin only or combined with something else | | | |
| Insulin only | 17.9 (17.5) | 14 (0–32) | 0.002 |
| Combined | 27.1 (16.9) | 25 (15–39.5) | |
| Concomitant disease | | | |
| No | 16 (13.4) | 15 (0–23) | 0.029 |
| Yes | 24.7 (18.1) | 25 (9.5–36) | |
| Fundoscopy | | | |
| Without findings | 16.1 (13.2) | 15.5 (2–26) | 0.002 |
| With findings | 26.6 (18.5) | 26.5 (11–39) | |
| Fundoscopy | | | |
| ● Without findings | 16.1 (13.2) | 15.5 (2–26) | **0.001 +** |
| Mild Non-Proliferative Diabetic Retinopathy | 17.5 (16.1) | 18 (0–32) | |
| Moderate Non-Proliferative Diabetic Retinopathy | 24.1 (14.5) | 23 (14–32) | |
| ● Severe Non-Proliferative Diabetic Retinopathy | 29.6 (15.4) | 31.5 (25–36) | |
| ● Proliferative Diabetic Retinopathy | 31.7 (21.0) | 30 (14–50) | |
| Diabetic Macular Edema | | | |
| No | 19.9 (16.1) | 20.5 (5.5–30) | 0.001 |
| Yes | 30.4 (18.9) | 32.5 (16–45) | |
| Treatment with anti-VEFG injections | | | |
| No | 20.5 (16) | 21 (6–30) | 0.002 |
| Yes | 32.3 (20) | 34 (16–49) | |
| Number of doses, r ‡ | −0.42 | | 0.015 |

+ Kruskal–Wallis test; ‡ Spearman's correlation coefficient; ● clinically significant differences between (1) without findings—Severe Non-Proliferative Diabetic Retinopathy—and (2) without findings—Proliferative Diabetic Retinopathy.

Multivariate analysis showed that having Severe Non-Proliferative Diabetic Retinopathy or having Proliferative Diabetic Retinopathy and receiving insulin as therapy (alone or in combination with another treatment), was significantly associated with greater anxiety symptoms (Table 4).

## 4. Discussion

This study evaluated the impact of Diabetic Retinopathy and diabetic macular edema, on vision-related quality of life, and investigated the potential association between disease severity, emotional status of patients, and the possible development of anxiety and psychological features. The data showed that the presence of findings in the fundoscopy was significantly associated with all VFQ-25 subscales (except for driving) as well as the composite score. More specifically, patients with Moderate Non-Proliferative Diabetic Retinopathy, Severe Non-Proliferative Diabetic Retinopathy or Proliferative Diabetic Retinopathy had significantly worse general health compared to patients without findings in their fundoscopy. This might reflect the fact that the more severe the ocular involvement in diabetes, the more increased the probability is of advanced disease in other organs. The aforementioned findings also underscore the significance of a correct therapeutic approach and glucose monitoring aimed at preventing DR development [19].

Furthermore, patients with Proliferative Diabetic Retinopathy seem to bear a heavier burden in many aspects including the general quality of life. They suffer from significantly greater anxiety and have significantly worse general health and vision, when compared

with patients without retinal lesions. The advanced nature of the disease is usually accompanied by general organic deterioration leading to further anxiety. In addition, this subgroup endorses more ocular pain and discomfort [4]. Proliferative disease was also found to be associated with worse visual performance, both for near and far activities, and impaired color and peripheral vision as well, further preventing patients from performing their daily life activities and cultivating anxiety. They also scored lower regarding social functioning, mental health, role adequacy and dependency indices. Not surprisingly, proliferative disease representing the advanced form of DR may be accompanied by neovascular glaucoma causing pain and discomfort, increased eye dryness and worse prognosis, having a detrimental impact on every aspect of the patient's life, both psychologically and functionally [13–15]. The PRP therapeutic approach is commonly performed in patients with Proliferative DR to deal with retinal ischemia, and neovascularization may also lead to ocular discomfort and impaired accommodation [12].

Another important factor having a negative impact and impairing the quality of life in DR patients is the presence of anxiety and its severity. Anxiety disorders have an annual prevalence of 14% at the ages of 14–65 and it is more common in women [21,22]. Regarding the prevalence of anxiety in adults with diabetes, a systematic review of the literature showed that generalized anxiety disorder is present in 14% and elevated symptoms of anxiety in 40% of patients with diabetes who participate in clinical studies [36]. GAD is of special interest, characterized clinically by alertness, and feelings of fear, tension and anxiety in dealing with daily life events [23,24].

Anxiety and Diabetic Retinopathy may represent interconnected entities with a potential association between the disease's severity, emotional status of patients and the development of anxiety and psychological disorders as well. Anxiety could also play a role in the worsening of DR. This study showed that greater anxiety symptoms were significantly associated with worse quality of life in all sectors except for general health, driving and color vision. Anxiety undermines patients' physical and psychological powers, rendering them less capable of performing tasks that patients with a disease of the same severity and without anxiety would be able to achieve. Anxiety also reflects on the psychological constituent that visual perception can have, with anxious patients underestimating their visual capability or responding under psychological burden [21–24]. Moreover, daily time of sitting seems to negatively affect general health, general vision, distance activities, social functioning, mental health and composite VFQ-25 score. Sitting may cultivate introversion and lead patients to resign from daily life activities including tasks involving vision. It is also associated with less time spent on working out, further impairing patients' general health [37,38].

In addition, employed patients had significantly higher scores in near activities, distance activities, social functioning, mental health, dependency and peripheral vision scales, indicating better quality of life. Employment has a positive impact on patients' psychology and seems to motivate them to upgrade their lives as much as they can and to improve their performance. Education was also found to contribute positively to patients' well-being [18]. Regarding "general vision", patients who were primary school graduates at most had significantly worse performance compared to patients who were college/university or MSc graduates. In general, patients with higher education tend to take better care of themselves and to be more alert regarding warning symptoms and signs of their visual function [18,19].

Not surprisingly, the presence of concomitant diseases, such as thyroid or heart disease was also associated with significantly greater anxiety. Additional comorbidities seem to constitute a psychological and emotional burden for patients with DR [19]. Moreover, anxiety was greater in those with findings in fundoscopy. Interestingly, the presence of diabetic macular edema, representing the main cause of visual deterioration in DR patients is found to be correlated with greater anxiety, as those patients have in general impaired vision. Treatment with anti-VEFG injections and the number doses of anti-VEFG injections were also significantly associated with greater anxiety. The thought of intraocular injections, especially when repeatedly needed, may scare patients and the injection itself is often accompanied by a certain amount of eye pain and discomfort.

Chronicity and severity of disease has its impact on emotional status. More years suffering with diabetes was significantly associated with greater anxiety. It may represent an increasing fear of progression in the course of disease or disappointment due to DR progression to more advanced stages. Advance stages of DR, that is, Severe Non-Productive Diabetic Retinopathy or Proliferative Diabetic Retinopathy, correspond to heavier disease burden with worse vision and, consequentially, greater anxiety. Also, being treated with insulin as a monotherapy or in combination was significantly associated with greater anxiety symptoms. Insulin use, the fear and pain of injections, and the deterioration in vision due to diabetic lesions constitute a psychological burden for patients [12]. In addition, sitting was associated with greater anxiety. Being active contributes to taking part in activities that increase the internal power against anxiety development [20]. Working out and levels of activity are crucial parameters of patients' well-being. Psychometrically adjusted questionnaires are useful tools in our armamentarium for timely recognition and intervention when needed.

Importantly, the potentially bi-directional relationship between Diabetic Retinopathy and anxiety is a crucial aspect of evaluating patients with diabetes and coexistent or developing disturbances in their emotional status or anxiety levels. Increased levels of circulating cytokines have characteristically been observed both in patients with diabetes and psychiatric disorders. It has been suggested that a combination of increased levels of cytokines and insulin deficiency results in neurocognitive deficits, inappropriate neural development and fluctuations in blood glucose [39–41]. Roy et al. (2007) [42] suggested that the HPA axis may play a major role in DR development involving the psychological constituent. Dysregulation in this pathway leads to hypercortisolemia and subsequently to alterations in insulin resistance, contributing to DR manifestation [42]. Although diabetes is an upstream event for stress, it may also be an outcome of chronic stress. Epidemiological studies have shown that diabetes is a common stress-driven disease and stressful life events increase the possibility of diabetes manifestation. Stress and insulin resistance are likely linked through molecular pathways, including pancreatic beta cells, lipid metabolism, the renin-angiotensin system, the autonomic nervous system, the immune response system and endocrine hormones which are under the influence of stress [43].

Diabetic Retinopathy and especially advanced stages of the disease, have a crucial impact on patients' quality of life and emotional status. Severely, visually impaired patients secondary to DR are facing difficulties in many fundamental aspects of their daily life and have to deal with insurmountable burdens, with adverse effects on their physical, mental and psychological well-being. Among others, they might have difficulty being independent and performing everyday tasks, such as reading the newspaper, working, cooking, sewing, going out, walking, driving, interacting socially, and taking care of themselves. In this context, emotional imbalance is a reasonable consequence, and psychological side effects and comorbidities, like anxiety disorders and even depression, could emerge, further deteriorating the patients' condition.

The findings of this study should be interpreted considering a number of limitations. First, as the assessment of patients was based only on self-report questionnaires, response and recall bias may have been introduced into this study. Second, this study was cross-sectional and therefore causal relationships cannot be established. Third, in the current study, the association between family history of diabetes and anxiety symptoms amongst DR patients was not examined and thus needs further investigation. One more limitation was that this study was conducted in one clinic, using convenience sampling and therefore the sample cannot be considered representative. Another limitation could be the small number of the study population.

This study can help us acquire a better understanding of the degree of social and emotional impact of DR, thus assisting policy planners, rehabilitation counselors and researchers in developing strategies for quality-of-life improvements in patients with DR. A timely diagnosis can positively affect the emotional and functional well-being of patients

in the long run. In addition, this study provides evidence-based reasoning regarding the necessity of psychological support to those diagnosed with this condition.

## 5. Conclusions

The well-established impact of DR in patients' well-being, quality of life and emotional status render DR and CME prevention, stabilization or delaying progression a necessity in order to protect patients from developing psychological diseases. On the other hand, the speculated bi-directional association between emotional problems and DR progression, brings to the forefront the importance of acknowledging and dealing with psychological issues aiming at delaying DR progression.

**Author Contributions:** Conceptualization, G.S.; methodology, V.E.; investigation, G.S. and D.R.; resources, G.S. and D.R.; writing—original draft, G.S.; writing—review and editing, E.H., V.E., P.T. and E.R.; supervision, E.H., P.T. and E.R. All authors have read and agreed to the published version of the manuscript.

**Funding:** This research received no external funding.

**Institutional Review Board Statement:** This study was conducted according to the guidelines of the Declaration of Helsinki and was approved by the Institutional Review Board—Scientific Council of 'Attikon' University General Hospital, Athens 12462, Greece, National and Kapodistrian University of Athens. Protocol code: (οφθ, ΕΒΔ2909/20-12-2017), date of approval: 31 January 2018.

**Informed Consent Statement:** Informed consent was obtained from all subjects involved in the study.

**Data Availability Statement:** Data is unavailable due to privacy or ethical restrictions.

**Conflicts of Interest:** The authors declare no conflict of interest.

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
