# Peer review of "Evaluation of Quality of Life and Emotional Disturbances in Patients with Diabetic Retinopathy"

_ejihpe, doi:10.3390/ejihpe13110175_

Round 1
Reviewer 1 Report
Comments and Suggestions for Authors
It is a topic of interest.
However,
Lines 97-100 : ,, All participants were requested to complete two self-report questionnaires, the first one reflecting their anxiety status; the Beck Anxiety Inventory (BAI) and, the second one, which is the National Eye Institute Visual Function Questionnaire-25 (NEI VFQ-25), which evaluates the vision-related Quality of Life (27-29).”
Is it just one rating? I ask this because in lines 110-112 you show that ,,The questionnaire can be given to the same patient in subsequent sessions to track the progression or improvement of the anxiety”
Lines 112-114: ,,The BAI discriminates effectively between anxious and non-anxious diagnostic groups and, as a result, it is useful as a screening measure for anxiety.”
Did you want to do a screening for anxiety in these patients?
Lines 140-143: ,,Initially, we evaluated the impact of DR on the BAI and VF-25 questionnaires based on the existence (first arm) or the absence (second arm) of funduscopic findings. Conse-quentially, subgroup analysis was performed, comparing the findings in the anxiety, and quality of life, scales among the 4 groups of different DR severity stage (29,34,35)”
In the Materials and methods chapter, you say that there were 5 groups (lines 79-83). So the quality of life was evaluated only in the study group? Without the control study?
Lines 295-296: ,,In addition, sitting was associated with greater anxiety.”
What are you referring to?
Lines 297-300: ,,Working out and levels of activity are crucial parameters of patients’ well-being. Psychometrically adjusted questionaries are useful tools in our armamentarium for timely recognition and intervention when needed.”
What exactly are you referring to? So far in the text you have not referred to these aspects.
Lines : 303-305: ,,Increased levels of circulating cytokines have characteristically been observed both in patients with diabetes and psychiatric disorders.”
So far in the text you have not made reference to cytokines. So, how were they determined and in what context? In addition to applying the questionnaires, did you also perform cytokine evaluation analyses?
It is to be appreciated that you also mention the limits of the study. If the study could have been carried out in more clinics and with a larger number of participants, perhaps we would have had more accurate data on the degree of social and emotional impact of diabetic retinopathy.
The Ethics Commission's opinion was issued in 2018. Perhaps it would be good to specify in the text the period during which this study took place.
For references:
No. 35 does not have the year of publication of the article. I searched and found 2013.
The references must be corrected, in some cases the year of publication of the article is placed after the authors, in others in parentheses, the full name and abbreviated first name rule is not followed, etc.
My comments are only intended to make the paper better. Good luck!
Reviewer 2 Report
Comments and Suggestions for Authors
Dear authors,
I find your review of interest. However I think that before publication the English needs to be improved. Furthermore, where the acronyms appear for the first time in the text, these should be written in full. Around line 248 there is a different character, it needs to be fixed.
Have you tried to make a gender distinction?
Comments on the Quality of English LanguageEnglish needs to be improved
Reviewer 3 Report
Comments and Suggestions for Authors
Dear authors,
Thank you for the opportunity to review the manuscript titled: “Evaluation of Quality of Life and Emotional Disturbances in Patients with Diabetic Retinopathy” for the EJIHPE journal.
The present work is interesting and relevant from a clinical point of view, because diabetes and its neuro-ophthalmological complications have an important influence on the psychoemotional and physical state of the patient, having a significant impact on physical and motor disorders.
However, I think an addition is necessary:
Page 1, section Abstract: The abstract of the paper is incomplete. Does not respect the style of structured abstracts. Not all the characteristic aspects are addressed: the purpose of the study, the methods applied, the results obtained and the conclusions.
Reviewer 4 Report
Comments and Suggestions for Authors
First of all I congratulate the authors for their meticulous efforts in putting patient data. Case studies like this are very important to bring the changes in the patient life style. I have only one suggestion in your manuscript.
1. In page 3, line 118, What is the full form of QOL? For the first time please write the full farm for the benefit of readers.
